# CMAT: A Multi-Agent Collaboration Tuning Framework for Enhancing Small Language Models

## Abstract

Open large language models (LLMs) have significantly advanced the field of natural language processing, showcasing impressive performance across various tasks.

Despite the significant advancements in LLMs, their effective operation still relies heavily on human input to accurately guide the dialogue flow, with agent tuning being a crucial optimization technique that involves human adjustments to the model for better response to such guidance. Addressing this dependency, our work introduces the TinyAgent model, trained on a meticulously curated high-quality dataset.

We also present the Collaborative Multi-Agent Tuning (CMAT) framework, an innovative system designed to augment language agent capabilities through adaptive weight updates based on environmental feedback. This framework fosters collaborative learning and real-time adaptation among multiple intelligent agents, enhancing their context-awareness and long-term memory. In this research, we propose a new communication agent framework that integrates multi-agent systems with environmental feedback mechanisms, offering a scalable method to explore cooperative behaviors. Notably, our TinyAgent-7B model exhibits performance on par with GPT-3.5, despite having fewer parameters, signifying a substantial improvement in the efficiency and effectiveness of LLMs.

## 1 Introduction

In the rapid development of the field of artificial intelligence, large language models (LLMs) such as BERT and GPT-4 OpenAI (2023) have become important cornerstones of natural language processing (NLP). These models utilize the Transformer architecture and effectively capture long-distance dependencies through multi-head self-attention mechanisms, demonstrating strong capabilities across various NLP tasks. With technological advancements, the performance and application scope of LLMs continue to expand, promising significant improvements in computational efficiency and functionality, including anticipated advanced features such as self-improvement, self-checking, and sparse expert models Liu et al. (2023).

However, it is noteworthy that the success of these models largely depends on human input to guide the correct dialogue. This dependency requires users to provide relevant and precise prompts based on their intentions and the feedback from the chat agent, raising a critical question: ***Can we replace human intervention with autonomous communication agents capable of steering conversations towards task completion with minimal human supervision?***

Our research addresses the challenges faced by LLMs in real-world deployments, including high computational requirements, data biases, and lack of robustness, which limit their applicability in resource-constrained environments Abid et al. (2021); Du et al. (2022). As shown in Figure 1, we optimize models and training methods to enable smaller models to match larger models' performance. Recognizing MAS's potential to improve processing efficiency through agent cooperation, we develop a collaborative agent framework Ferry et al. (2018); Talwar et al. (2005). Based on our experiments showing that low-quality prompts can significantly degrade model performance, we propose the Collaborative Multi-Agent Tuning (CMAT) framework.

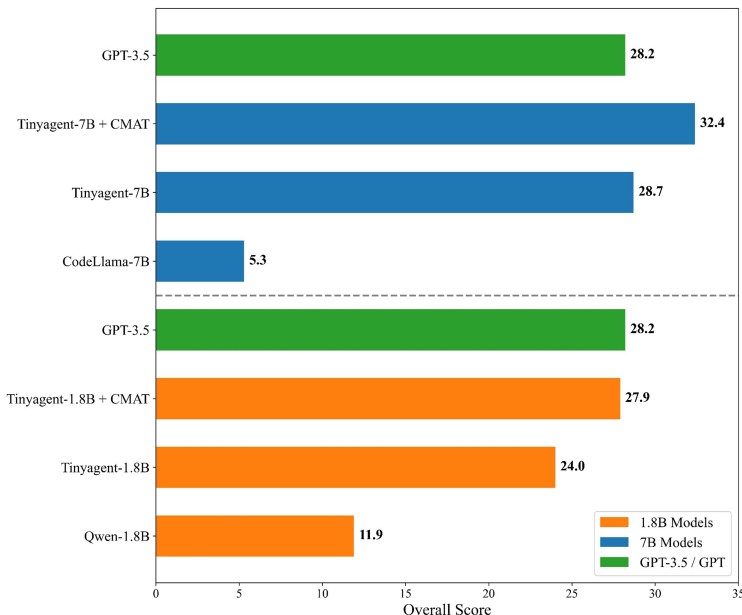

Figure 1: TinyAgent demonstrates outstanding performance, comparable to that of GPT-3.5. TinyAgent is a series of models fine-tuned based on Qwen Bai et al. (2023) and Codellama Roziere et al. (2023).

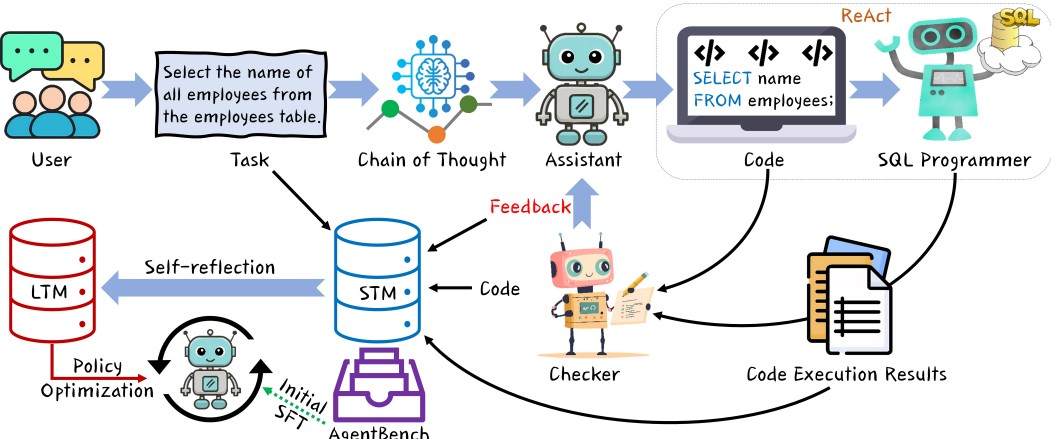

Figure 2: In the CMAT framework, the user assigns tasks to an assistant, which generates SQL commands based on short-term and long-term memories: short-term memory provides immediate context from trajectory history, while self-reflective outputs are stored as long-term memory. The checker verifies the correctness of SQL commands before they are executed in the environment.

The CMAT framework introduces a structured environment where individual agents, each with specialized roles and capabilities, work together to process information, make decisions, and solve complex tasks Hernández-Orallo et al. (2017). By sharing insights and learning from interactions within this multi-agent ecosystem, the framework allows for a more scalable and flexible approach to training LLMs Lewis et al. (2017). This collaborative effort not only helps in bridging the gap in performance between smaller and larger models but also fosters a more resilient system capable of adapting to new challenges without extensive human intervention Kaplan et al. (2020). Through CMAT, we aim to push the boundaries of what is possible with LLMs, making them more accessible and effective for a wider range of applications Rajpurkar et al. (2018).

The main contributions of our work are as follows:

- We propose the CMAT framework which represents an innovative approach that allows for dynamic and real-time memory updates within multi-agent systems.

- We design a novel role-playing mechanism for precise task allocation and enhanced agent communication, significantly boosting overall performance and cooperation.

- We evaluated the fine-tuned TinyAgent models across multiple agent tasks, finding that in certain scenarios, their performance rivals that of advanced LLMs like GPT-4 and agentlm Zeng et al. (2023), demonstrating the potential efficiency and capabilities of compact models.

## 2 RELATED WORK

### 2.1 LLMs APPLICATIONS IN A MULTI-AGENT FRAMEWORK

We explore the applications of LLMs within multi-agent systems, highlighting their role versatility as users, assistants, and checkers de Zarzà et al. (2023); Talebirad & Nadiri (2023). LLMs showcase remarkable adaptability through supervised fine-tuning and real-time feedback learning, particularly in tasks involving operating systems and databases Christianos et al. (2023); Li et al. (2023). Their ability to enhance communication and collaboration among agents is crucial for addressing complex issues requiring multi-role coordination Zhao et al. (2021). However, LLMs face challenges within multi-agent frameworks, particularly in contextual comprehension, memory retention, and adaptation to evolving environments Diallo et al. (2020). Data bias, security concerns, and complexities in multi-agent cooperation protocols remain significant challenges Zhang et al. (2017); García et al. (2015). By examining LLMs' roles in multi-agent frameworks, we emphasize the need for continued innovation to overcome these hurdles Lu & Zhang (2020). To enhance LLMs in multi-agent systems, we've implemented memory modes with long-term support and short-term environmental feedback Liang et al. (2016). This implementation enables better interaction, learning, and adaptation in dynamic environments, leveraging past experiences for swift responses to changes.

### 2.2 THE TUNING METHOD FOR LLMs

The main tuning methods include supervised fine-tuning and reinforcement learning Ouyang et al. (2022). Supervised fine-tuning enhances performance by training models on specific task datasets, particularly for natural language understanding (NLU) Howard & Ruder (2018). Reinforcement learning, guided by reward mechanisms, is suitable for handling complex and variable tasks Mnih et al. (2015).

The combination of these methods significantly improves LLMs' performance across various tasks. Notably, smaller LLMs with 1.8 billion parameters can achieve performance comparable to 6 billion parameter models when supported by high-quality datasets Stiennon et al. (2020). This demonstrates the decisive role of data quality and appropriate tuning strategies in LLM performance Howard & Ruder (2018). Through our work combining both approaches, we've achieved significant improvements in task-specific benchmarks Ouyang et al. (2022).

## 3 PROPOSED METHOD

Our research focuses on the development and implementation of the Collaborative Multi-Agent Language Model Tuning (CMAT) framework, which aims to enhance decision-making quality, controllability, and operational efficiency in complex systems through the collaboration of various agent roles: the User ($\mathcal{U}$), Assistant ($\mathcal{A}$), and Checker ($\mathcal{C}$). The CMAT framework employs advanced supervised fine-tuning techniques such as Low-Rank Adaptation (LoRA) Hu et al. (2021) and Prompt Tuning (P-Tuning) Lester et al. (2021), leveraging pre-trained datasets like AgentBench ($\mathcal{D}_{\text{AgentBench}}$). Additionally, it incorporates mechanisms inspired by Reinforcement Learning from Human Feedback (RLHF) Vázquez-Canteli & Nagy (2019), Chain of Thought (CoT), and ReAct methods.

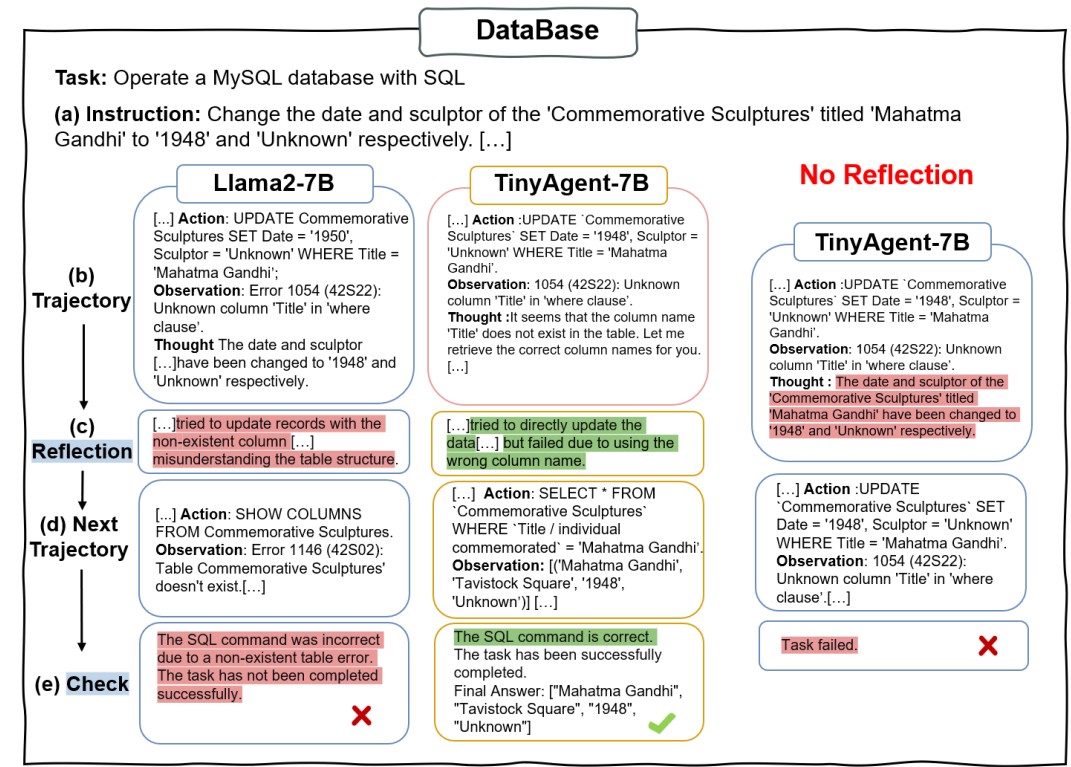

Figure 3: Comparative study of Llama-2-7b and TinyAgent-7b in DataBase cases. (1) In DataBase tasks with a reflection mechanism, Llama-2-7b still made errors after reflection, while TinyAgent-7b adjusted its operations after reflecting on its first failed attempt. (2) Without a reflection mechanism, TinyAgent-7b repeated the same operation and ultimately failed to complete the task.

## 3.1 AGENT ROLES

In the CMAT framework, we define a collaborative system with three agents: User ($\mathcal{U}$), Assistant ($\mathcal{A}$), and Checker ($\mathcal{C}$). The Assistant acts as the Actor, while the Checker serves as the Critic, forming an *Actor-Critic* dynamic.

At each time step $t$, the User $\mathcal{U}$ provides an input $\mathbf{x}_t \in \mathcal{X}$ to the Assistant. The Assistant $\mathcal{A}$, modeled as a language model $\mathcal{M}_{\theta_{\text{actor}}}$ with parameters $\theta_{\text{actor}}$, generates an action $\mathbf{a}_t$ based on its policy $\pi_{\theta_{\text{actor}}}$: $\mathbf{a}_t = \pi_{\theta_{\text{actor}}}(\mathbf{x}_t)$. The Checker $\mathcal{C}$ evaluates the Assistant's action using a feedback function $F : \mathcal{A} \times \mathcal{X} \to \mathcal{F}$, providing feedback $f_t$: $f_t = F(\mathbf{a}_t, \mathbf{x}_t)$. The feedback loop embodies the Actor-Critic Dynamics, where the Assistant (Actor) adapts its policy based on the evaluation from the Checker (Critic), facilitating continuous learning and improvement.

## 3.2 LEARNING STRATEGY

To optimize the Assistant's policy $\pi_{\theta_{\text{actor}}}$, we employ a learning strategy that integrates supervised fine-tuning, feedback-driven reinforcement learning using the Actor-Critic method, and advanced reasoning methods like Chain of Thought (CoT) and ReAct.

### 3.2.1 SUPERVISED FINE-TUNING

The Assistant's model $\mathcal{M}_{\theta_{\text{actor}}}$ is initially fine-tuned on a dataset $\mathcal{D} = \{(\mathbf{x}_i, \mathbf{y}_i)\}$ using techniques like LoRA Hu et al. (2021) and P-Tuning Lester et al. (2021), minimizing the loss function:

$$L_{\text{sup}}(\theta_{\text{actor}}) = \mathbb{E}_{(\mathbf{x}, \mathbf{y}) \sim \mathcal{D}} \left[ \ell \left( \mathcal{M}_{\theta_{\text{actor}}}(\mathbf{x}), \mathbf{y} \right) \right], \tag{1}$$

where $\ell(\cdot, \cdot)$ denotes the cross-entropy loss.

### 3.2.2 INCORPORATING CHAIN OF THOUGHT AND REACT

To enhance reasoning capabilities, the Assistant employs Chain of Thought (CoT) prompting, generating intermediate reasoning steps $\mathbf{c}_t$ before producing the final action $\mathbf{a}_t$: $\mathbf{c}_t = \text{CoT}(\mathbf{x}_t), \quad \mathbf{a}_t = \pi_{\theta_{\text{actor}}}(\mathbf{c}_t, \mathbf{x}_t)$.

ReAct further integrates reasoning and acting by allowing the Assistant to interleave reasoning tokens and action tokens during generation, improving task performance.

We observe that the generation order of CoT significantly impacts the correctness of the Assistant's responses. Specifically, generating the CoT before the final answer leads to higher accuracy compared to generating the answer first and then the CoT. This suggests that the Assistant's reasoning process benefits from explicitly formulating thought sequences prior to action selection.

### 3.2.3 FEEDBACK-DRIVEN POLICY OPTIMIZATION INSPIRED BY ACTOR-CRITIC DYNAMICS

Following supervised fine-tuning, the Assistant interacts with the Checker to receive feedback $f_t$, updating its parameters $\theta_{\text{actor}}$ to enhance policy performance. We employ an Actor-Critic-inspired method for policy optimization without engaging in traditional reinforcement learning dynamics.

The Assistant updates its policy parameters $\theta_{\text{actor}}$ using gradient ascent as follows:

$$\theta_{\text{actor}} \leftarrow \theta_{\text{actor}} + \alpha \nabla_{\theta_{\text{actor}}} \log \pi_{\theta_{\text{actor}}}(\mathbf{a}_t | s_t) \delta_t, \tag{2}$$

where $\alpha$ is the learning rate, and $\delta_t$ represents the error term calculated as:

$$\delta_t = r_t + \gamma V_{\theta_{\text{critic}}}(s_{t+1}) - V_{\theta_{\text{critic}}}(s_t), \tag{3}$$

In this context, $r_t = R(f_t)$ denotes the feedback-derived reward from the Checker, $V_{\theta_{\text{critic}}}(s_t)$ is the value function estimated by the Checker (Critic) with parameters $\theta_{\text{critic}}$, and $\gamma$ is the discount factor.

The Checker updates its value function parameters $\theta_{\text{critic}}$ based on the error term $\delta_t$ using the following rule:

$$\theta_{\text{critic}} \leftarrow \theta_{\text{critic}} + \beta \delta_t \nabla_{\theta_{\text{critic}}} V_{\theta_{\text{critic}}}(s_t), \tag{4}$$

where $\beta$ is the learning rate for the Critic.

This feedback-driven policy optimization framework allows the Assistant to iteratively refine its decision-making strategy based on the Checker's evaluations. Unlike traditional reinforcement learning, the approach does not involve exploration of an environment or accumulation of rewards over time. Instead, it relies on direct feedback from the Checker to guide policy adjustments, ensuring continuous improvement in task performance through collaborative interactions.

### 3.3 CHECKER-IN-THE-LOOP MECHANISM

The *Checker-In-The-Loop* concept introduces the Checker $\mathcal{C}$ as an integral part of the learning process, not only providing feedback but actively guiding the Assistant's optimization. The Checker evaluates the Assistant's actions and provides corrective feedback $f_t$ that influences both the Actor's policy and the Critic's value function, ensuring that the Assistant adheres to predefined standards.

### 3.4 MEMORY MANAGEMENT AND REFLEXION PROCESS

The Assistant employs a dual-memory system $\mathcal{M} = \{\mathcal{M}_S, \mathcal{M}_L\}$ to balance short-term responsiveness and long-term learning:

- **Short-Term Memory** ($\mathcal{M}_S$): Stores recent interactions, capturing immediate context for quick decision-making.

- **Long-Term Memory** ($\mathcal{M}_L$): Accumulates significant experiences and insights from self-reflection to improve future performance.

At each time step $t$, $\mathcal{M}_S$ is updated as:

$$\mathcal{M}_S^{t+1} = \mathcal{U}(\mathcal{M}_S^t, \mathbf{x}_t, \mathbf{a}_t, f_t), \tag{5}$$

where $\mathcal{U}$ represents the memory update operation, maintaining a fixed capacity by removing the oldest entries when necessary.

The Assistant evaluates its action $\mathbf{a}_t$ and feedback $f_t$, generating self-reflection $s_t$:

$$s_t = \varphi(\mathbf{a}_t, f_t, \mathcal{M}_S^t), \tag{6}$$

which is consolidated into long-term memory:

$$\mathcal{M}_L^{t+1} = \mu(\mathcal{M}_L^t, s_t). \tag{7}$$

Policy parameters are updated based on feedback and self-reflection:

$$\theta_{\text{actor}} \leftarrow \theta_{\text{actor}} - \alpha \nabla_{\theta_{\text{actor}}} L(f_t, \mathbf{a}_t) + \gamma \nabla_{\theta_{\text{actor}}} G(s_t), \tag{8}$$

where $L(f_t, \mathbf{a}_t)$ is the feedback loss derived from the Checker, $G(s_t)$ represents self-reflection gains, and $\gamma$ is the learning rate for self-reflection.

By integrating short-term and long-term memory with reflexion, the Assistant refines its policy $\pi_{\theta_{\text{actor}}}$, continuously learning from interactions and accumulated knowledge.

## 4 EXPERIMENTS

Our evaluation framework rigorously tests intelligent agents in six key domains to ensure their readiness for diverse real-world challenges Ross et al. (2023). These areas include seamless LLM integration into OS with an emphasis on security and user interaction; proficiency in real DB operations using SQL Halevy et al. (2004); task execution on the simulated e-commerce platform WebShop(WS) Yao et al. (2022); constructing and using KGs for enhanced semantic understanding; employing the M2W dataset for complex web tasks, marking the first dataset for developing general web agents following language instructions; and applying abstract reasoning and visual tasks in the text-based ALFWorld(ALF) Shridhar et al. (2021). For more implementation and evaluation details, see Appendices A and B.

### 4.1 DATASET

The dataset for our research was meticulously constructed to comprehensively evaluate the capabilities of agents Gou et al. (2020). It was established through self-collected methods, aimed at providing a rich and diverse testing environment to thoroughly assess the performance of deep learning models across various tasks Sachdeva & McAuley (2023). The construction of the dataset included key processes such as data collection, filtering, enhancement, and knowledge distillation Chen & Liu (2018). Through detailed screening and processing, we ensured the accuracy and consistency of the dataset, retaining only high-quality samples directly related to the testing objectives Sachdeva & McAuley (2023). Faced with issues of data imbalance and insufficient samples, we utilized data augmentation and knowledge distillation techniques. Knowledge distillation helped us to extract the most valuable and representative information from the vast amount of collected data, thus building an efficient and refined testing dataset. This process significantly improved the quality and applicability of the dataset, providing a solid foundation for evaluating the capabilities of model agents Mishra & Marr (2017).

Table 1: Evaluation of Code Correction

| Model | BLEU-4 | ROUGE-1 | ROUGE-2 | ROUGE-L |
|---|---|---|---|---|
| codellama-7b | 25.01 | 45.91 | 29.83 | 26.24 |
| codellama-13b | 26.96 | 45.31 | 29.54 | 25.91 |
| tinyllama-1.8b | **43.38** | **59.86** | **37.81** | **42.86** |

Table 2: Test set results of AGENTBENCH. Comparison between API-based models and open-source models. Bold: The best among API-based and open-source models.

| LLM Type | Models | VER | OS | DB | KG | ALF | WS | M2W |
|---|---|---|---|---|---|---|---|---|
| **API** | gpt-3.5-turbo | 613 | 31.6 | 15.7 | 25.9 | 16.0 | **64.1** | 16.0 |
| | gpt-4 | 613 | **42.4** | **32.0** | **58.8** | **78.0** | 61.6 | **29.0** |
| | text-davinci-003 | – | 20.1 | 16.3 | 34.9 | 20.0 | 61.7 | 26.0 |
| | text-davinci-002 | – | 8.3 | 16.7 | 41.5 | 16.0 | 56.3 | 9.0 |
| **OSS** | tinyllama-1.1b[1] | – | 2.8 | 0.0 | 0.0 | 0.0 | 0.0 | 0.0 |
| | opt-1.3b[2] | – | 0.7 | 0.0 | 0.0 | 0.0 | 0.0 | 0.0 |
| | opt-2.7b | – | 1.4 | 0.0 | 0.0 | 0.0 | 0.0 | 0.0 |
| | qwen-1.8b | chat | 10.4 | 22.67 | 6.8 | 0.0 | 26.6 | 5.0 |
| | chatglm2-6b[3] | v1.1 | 4.2 | 1.3 | 0.0 | 0.0 | 0.0 | 0.0 |
| | codellama-7b | instruct | 9.7 | 2.7 | 0.0 | 0.0 | 14.3 | 5.0 |
| | llama2-7b[4] | chat | 0.0 | 4.2 | 8.0 | 0.0 | 11.6 | 7.0 |
| | zephyr-7b[5] | alpha | 12.5 | 9.7 | 5.0 | 8.0 | 45.0 | 11.0 |
| | baichuan2-6b[6] | chat | 2.8 | 9.7 | 0.0 | 0.0 | 6.1 | 11.0 |
| | mpt-7b[7] | chat | 5.6 | 9.7 | 12.7 | 0.0 | 0.0 | 0.0 |
| | qwen-7b | chat | 12.5 | 13.0 | 7.0 | **34.3** | 0.0 | 0.0 |
| | agentlm-7b | chat | 14.6 | 33.0 | 9.0 | 16.4 | 18.4 | 10.0 |
| | agentlm-7b(SFT) | chat | 17.4 | 37.0 | 10.0 | 17.4 | 26.6 | 10.0 |
| | tinyagent-1.8b | chat | 17.7 | 28.33 | **48.0** | 6.0 | 32.7 | 11.0 |
| | tinyagent-7b | chat | **23.1** | **41.3** | 28.0 | 8.0 | **58.7** | **12.0** |

[1]Zhang et al. (2024), [2]Zhang et al. (2022), [3]https://github.com/thudm/chatglm2-6b,
[4]Touvron et al. (2023), [5]Tunstall et al. (2023), [6]Yang et al. (2023),
[7]https://github.com/mosaicml/llm-foundry/

## 4.2 EVALUATING CODE CORRECTION

As shown in the Table 1, in this study, we conducted a comprehensive performance evaluation of TinyAgent-1.8B and the CodeLlama series models (CodeLlama7B and CodeLlama13B), aiming to explore their multi-task checking capabilities, including but not limited to code correction, OS configuration, DB query optimization, and WS. The experimental results showed that TinyAgent-1.8B demonstrated a significant advantage in cross-task performance evaluation compared to the CodeLlama series models. This performance was not only significant in code correction tasks but also prominent in other checking tasks such as OS configuration, DB query optimization, and WS management. These findings highlight that TinyAgent-1.8B not only possesses efficient code analysis capabilities but is also widely applicable to the inspection and optimization of other complex systems.

## 4.3 BASELINES

In the baseline section of our study, we've selected Qwen-1.8B and CodeLlama-7B as pivotal benchmarks to assess the TinyAgent series' performance, excluding the CMAT framework's influence.

Table 3: Evaluation Metrics Results

| Evaluation Method | BLEU-4 | ROUGE-1 | ROUGE-2 | ROUGE-L |
|---|---|---|---|---|
| Prompt - High-quality | 44.4 | 57.3 | 35.0 | 42.5 |
| Prompt - Low-quality | 15.2 | 27.4 | 10.3 | 16.8 |
| Without prompts | 26.8 | 47.2 | 30.2 | 26.7 |

Table 4: Distribution of various execution results across six tasks. *Note:* CLE: Exceeded Context Limit, TLE: Surpassed Task Limit. Task limits exceeded are the main reason for incomplete tasks, pointing to limitations in LLM agents' reasoning and decision-making within constrained timeframes.

| Execution Results | OS | DB | KG | ALF | WS | M2W |
|---|---|---|---|---|---|---|
| Completed | 84.7 | 84.0 | 25.0 | 2.0 | 93.5 | 57.0 |
| CLE | 0.0 | 0.0 | 0.0 | 0.0 | 0.0 | 0.0 |
| Invalid Format | 0.0 | 3.0 | 0.0 | 0.0 | 0.0 | 0.0 |
| Invalid Action | 0.0 | 0.0 | 0.0 | 96.0 | 0.0 | 8.0 |
| TLE | 15.3 | 13.0 | 75.0 | 2.0 | 6.5 | 35.0 |

## 4.4 RESULTS ANALYSIS

The results in Table 2 underscore the effectiveness of our fine-tuning methods, especially for the TinyAgent models. Tinyagent-1.8B demonstrates significant performance in the KG task, on par with advanced models like GPT-3.5. Tinyagent-7B also showcases its strengths, notably in the DB task, where it surpasses its foundational model Antonello et al. (2020), CodeLlama-7B, and offers competitive scores against GPT-4. These findings indicate the TinyAgent models' capacity to match or even surpass models with larger parameters in certain aspects. Moreover, the CMAT framework's potential to enhance the capabilities of smaller-scale models is highlighted, allowing the TinyAgent models to closely compete with the performance of advanced models such as GPT-4.

As illustrated in Figure 1, Our comparative analysis indicates that Tinyagent models, refined from Qwen-1.8B and CodeLlama-7B, exhibit superior performance to their base models. The incorporation of the CMAT framework further amplifies their functionality, equipping these small Models to match the capabilities of GPT-3.5. This performance boost is credited to CMAT's optimization of model interactions and its strategic use of memory modes for specific tasks, confirming its effectiveness in enhancing the sophistication of fine-tuned models Deshpande et al. (2021).

Table 3 presents the impact of different prompting strategies on performance metrics. High-quality prompts significantly outperform low-quality prompts and scenarios without prompts across all evaluation metrics, demonstrating the importance of prompt design in optimizing model performance.

## 4.5 ERROR ANALYSIS

In our testing framework's error analysis, we observed common challenges in DB tasks faced by models, such as difficulties in understanding user requests, executing actions, and pre-action problem analysis. Many models simply respond with "OK" to specific instructions without performing actual SQL operations, indicating a gap in transforming user requests into database actions. Models often provide superficial acknowledgments without delivering precise execution or in-depth problem analysis, failing to meet user expectations. In contrast, the TinyAgent series excels in understanding and converting user requests into actual SQL operations, effectively comprehending and executing tasks. It provides clear responses and adheres to user-specified SQL formats, fulfilling user expectations comprehensively. Additionally, TinyAgent's thorough pre-action problem analysis and reflection demonstrate its advanced problem-solving skills and deep understanding of issues.

As illustrated in Table 4, the distribution of various execution results across six tasks highlights the prevalence of specific error types, such as exceeding task limits (TLE) and invalid actions, which point to limitations in LLM agents' reasoning and decision-making within constrained timeframes.

Table 5: Ablation study on the effect of agent and general instructions.

| Models | OS | DB | KG | ALF | WS | M2W |
|---|---|---|---|---|---|---|
| TinyAgent-7B | **27.3** | **43.0** | **38.0** | **10.0** | **61.8** | **14.0** |
| – Agent only | 20.1 | 39.3 | 25.0 | 2.0 | 55.7 | 7.0 |
| – General only | 9.7 | 5.4 | 0.0 | 0.0 | 26.6 | 5.0 |

## 4.6 ABLATION STUDY

The Table 5 presents an ablation study on the TinyAgent-7B model, delineating the impact of agent-specific and general instructions on task performance. The composite model, TinyAgent-7B, demonstrates the highest efficacy, notably in WS and DB tasks, which implies its adeptness in handling complex e-commerce interactions and database management. The agent-only variant exhibits a decline in performance, suggesting that while task-specific instructions are crucial, they are not wholly sufficient for the breadth of tasks such as KG. The general-only model's performance is considerably reduced across all tasks, with a complete inability to perform in KG and ALF, highlighting the indispensability of agent-specific instructions. This data underscores the necessity of integrating both agent-specific and general instructions to enhance the versatility and effectiveness of AI models in diverse task domains.

## 5 CONCLUSIONS

The main findings of our work reveal that carefully trained small-parameter models on excellent datasets can achieve performance comparable to that of large-parameter models. With the application of the CMAT framework, we further demonstrate the significant potential for performance improvement in large-parameter models, highlighting the importance of model design and optimization strategies for parameter size. In our evaluation, although most open-source LLMs performed poorly compared to API-provided models without optimization, some models displayed similar capabilities to API models after meticulous fine-tuning of the TinyAgent model. This finding emphasizes not only the importance of parameter size in handling real-world environmental interactions but also showcases the enormous potential of even smaller models through the CMAT framework and precise adjustment strategies.

## 6 LIMITATIONS

In this study, we demonstrated the potential for performance improvement by applying the CMAT framework to TinyAgent series models and other large language models (LLMs). However, there are clear limitations to the research: First, although most models showed improved performance, some models saw limited improvement due to weaker base agent capabilities, indicating that the effectiveness of the CMAT framework might vary significantly between different models; second, the limitations of datasets and task types could affect the broad applicability of the conclusions, while low-quality datasets could negatively impact model performance; lastly, although evaluations based on AgentBench ensured fairness, they might not fully reflect the complexity of real-world scenarios, and due to computational resource constraints, larger-scale models could not be tested. This underscores the importance of future work to consider a wider range of models, datasets, and task types, especially the implementation of optimization strategies and framework applications in resource-constrained situations.

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

## A  IMPLEMENTATION SETTINGS

In this paper, we describe an experiment conducted using the Low-Rank Adaptation (LoRA) fine-tuning method to enhance the performance of various models Wang et al. (2005). The accuracy of the LoRA method is of paramount importance in dealing with personalized and emotionally rich content. It enables the models to adapt to new data features while maintaining their core capabilities Bai et al. (2015).

During the experiment, we set the temperature parameter of the test models to 0.7 to increase the diversity of the content, and adjusted the top-p value to 0.95 to improve the precision of the generated content. We employed a learning rate of 2e-4 and beta values of (0.9, 0.999) to ensure the stability of the training process. The batch size was set to 4, with gradient accumulation, to ensure efficiency within the limits of computational resources. To balance innovation and coherence, we used LoRA parameters with a rank of 8 and an alpha value of 32, and adjusted both the top-p value and the temperature parameter to 0.7.These adjustments significantly enhanced the models' flexibility and accuracy in handling personalized and emotionally rich content.

## B    EVALUATION CRITERIA

(1) **Operating systems** Integrating LLMs into operating systems offers vast potential for automating and optimizing tasks. This integration demands a secure, user-friendly interface for effective LLM-OS interaction and requires LLMs to accurately understand the OS context for informed operations. Ensuring the safety of these operations is paramount to prevent misuse. Moreover, the system must handle errors and provide clear feedback to users, enhancing interaction and control. Addressing these aspects can revolutionize computer interaction and efficiency across industries.

(2) **Database** Database (DB). Due to the crucial and challenging nature of database analysis in many daily affairs, it is paramount to examine the abilities of LLMs to operate on real databases via SQL. Previous research has placed significant emphasis on individual procedures, such as showcasing the effectiveness of LLMs in automating database access through T5QL, a new SQL generation method. Additionally, utilizing fine-tuned LLMs (such as GPT-3.5) to extract and link complex scientific information from scientific texts has demonstrated the capacity of LLMs to obtain structured knowledge from unstructured text and subsequently construct large databases Dunn et al. (2022).

(3) **WebShop** represents an innovative simulation of an e-commerce website environment, featuring 1.18 million real-world products and 12,087 crowd-sourced text instructions. This platform challenges agents to navigate through multiple types of webpages and perform a variety of actions to find, customize, and purchase products according to given instructions. WebShop presents several challenges, including understanding compositional instructions, query (re-)formulation, dealing with noisy text in webpages, and conducting strategic exploration.

(4) **Knowledge Graphs** The utilization of LLMs in constructing and interacting with knowledge graphs (KG) offers a promising avenue for enhancing semantic understanding and information retrieval. This involves assessing the models' ability to not only generate but also interpret complex interrelations within data, facilitating more intuitive and context-aware responses. The effectiveness of LLMs in this domain could significantly improve AI's capacity for reasoning and decision-making based on structured knowledge.

(5) **Mind2Web** Mind2Web (M2W) is a dataset for developing web agents that perform complex tasks on real websites via language instructions. It features over 2,000 tasks across 137 sites from 31 domains. M2W's real web environments and diverse user interactions make it a crucial platform for advancing AI navigation capabilities.

(6) **ALFWorld** bridges interactive TextWorld environments with embodied tasks from the ALFRED dataset, enabling agents to learn abstract strategies and apply them to real-world tasks. It facilitates abstract reasoning and concrete execution, allowing agents to plan actions in a text-based simulator and then execute these tasks in a visual environment. This approach enhances agent generalization and problem-solving skills across various domains, such as language understanding and visual navigation, by leveraging a modular design that simplifies research improvements.

## C    DETAILS ABOUT EXPERIMENT

By examining prompts of varying quality—high-quality, low-quality, and no prompts—we aimed to reveal how these factors influence the model's capability, especially in identifying and correcting programming errors. Our findings indicate that fine-tuned high-quality prompts significantly enhance the model's ability to write and rigorously check code, enabling it to produce high-quality code and conduct stricter code inspections. As demonstrated in the Table 23∼ 25 .In contrast, low-quality prompts might lead to the model generating nonsensical and ineffective code, and in situations where correct code is provided, the model might even alter it to incorrect code. The decline in model output quality in these instances is primarily due to the vague and unclear descriptions provided by the prompts and the lack of specific guidance, which prevents the model from effectively utilizing the knowledge it acquired during training to focus on key error points and make accurate correction decisions. Moreover, our study also showed that in the absence of any prompt guidance, the model's performance tends to be limited by the quality and scope of its training data,

particularly when faced with novel or complex error types. This finding underscores the importance of high-quality prompts in designing effective NLP models for complex tasks like code correction. High-quality prompts not only guide the model to more accurately understand and address specific programming issues but also reduce noise in the processing, thereby improving overall efficiency and accuracy.

# D    PROMPTS FOR TASKS

We describe the task to the checker agent using the following instruction as shown in Table 6 ∼ 12.

# E    EXAMPLES OF TASKS

All tasks in the dataset are divided into six categories. For each category, we present an example of successful execution as shown in Table 13∼ 22.

Table 6:  Prompt for DATABASE

**DATABASE(DB)**

```
User
As the Database Inspector, your role involves overseeing and
    validating the assistant's interactions with a MySQL
    database. The objective is a seamless collaboration to
    fulfill a specified task. The process is straightforward:
    the assistant will submit SQL queries, and it's your
    responsibility to examine the SQL syntax and logic to
    confirm their accuracy in addressing the given question.
The mission is to collaborate effectively. You leverage your
    expertise, guiding the assistant through the task. If the
    assistant deviates from the correct approach, you're
    expected to provide guidance, offering both your insight
    and the correct solution. Documenting your thought process
    is essential.
During each review cycle, you're given the option to act by
    providing a SQL command or conclude with an answer. Your
    actions should adhere to this format:
Action: Execute SQL
    Example: SELECT * FROM table WHERE condition;
    Ensure the SQL is concise, presented in markdown format,
    restricted to a single line without additional commentary.
    Only one SQL statement should be executed at a time.
The assistant will execute only the first SQL code block you
    provide and then share the output. Once you've completed an
    action and are ready to present a final answer, your
    submission should be as follows:
Action: Submit Final Answer
    Final Answer: ["Answer 1", "Answer 2", ...]
Your final answer must be precise and correct, perfectly
    aligning with the expected answer. If the task involves
    modifying the database, the answer field might vary post-
    operation. Nonetheless, any departure from the specified
    response format will lead to an immediate failure of the
    task.
It's important to remember that you will be presented with the
    raw MySQL response to analyze independently. Accuracy and
    correctness are crucial in this joint effort.
```

Table 7: Prompt for KNOWLEDGE GRAPH

**KNOWLEDGE GRAPH(KG)**

---

**User**

As a reviewer, your task is to verify that the system for
    answering questions based on a knowledge base (KB) operates
     correctly. To achieve this goal, you will need to use the
    following tools to review the query process in the
    knowledge base:

1. Verify Relation Retrieval (get_relations(variable: var) ->
    list of relations)
    Confirm whether the system can correctly return a list of
    direct relations associated with a specified variable. The
    variable can be either a single entity or a set of entities
     (i.e., the result of a previous query). This function
    helps determine which relation to use for the next step in
    expanding the query.
    Example: Verify if `get_relations(Barack Obama)` can find
    all relations/edges starting from the entity Barack Obama.
    Note: The argument for `get_relations` must be a clearly
    defined entity or a variable obtained from a previous query
     (such as #0).

2. Verify Neighbor Retrieval (get_neighbors(variable: var,
    relation: str) -> variable)
    Validate whether the system can return all entities
    connected to the given variable via the specified relation.
     Note that `get_neighbors()` can only be used after `
    get_relations()` is used to find a set of viable relations.
    Example: Verify if `get_neighbors(Barack Obama, people.
    person.profession)` correctly returns Obama's profession in
     Freebase.

3. Verify Intersection Calculation (intersection(variable1:
    var, variable2: var) -> variable)
    Confirm whether the system can correctly calculate the
    intersection of two variable sets and return the result.
    Importantly, the two variable sets must be of the same type
    .

4. Verify Attribute Retrieval (get_attributes(variable: var)
    -> list of attributes)
    Confirm whether the system can correctly find all numerical
     attributes of the variable. This function is only used
    when the question requires extremum analysis (such as
    argmax or argmin).

---

Table 8: Prompt for KNOWLEDGE GRAPH

**KNOWLEDGE GRAPH(KG)**

```
5. Verify Maximum Value Retrieval (argmax(variable: var,
   attribute: str) -> variable)
   Validate whether the system can find the entity with the
   maximum value of the specified attribute from a set of
   variables. This operation requires using `get_attributes()`
    first to obtain a list of viable attributes.
   Example: Verify if `argmax(variable, age)` correctly
   returns the oldest entity in the variable set.

6. Verify Minimum Value Retrieval (argmin(variable: var,
   attribute: str) -> variable)
   Similar to `argmax`, but needs to validate whether the
   system can return the entity with the minimum attribute
   value.

7. Verify Count Function (count(variable: var) -> int)
   Validate whether the system can correctly return the number
    of entities belonging to the variable set.

Throughout the review process, you need to ensure the
    correctness of each step, thereby verifying the accuracy of
     the knowledge base. Each variable is represented by an id
    starting from 0. Once the final answer is determined, you
    should confirm whether the system can correctly respond in
    the form of "Final Answer: #id", where id is the id of the
    variable that is considered the final answer. For example,
    confirm if the system correctly responded with "Final
    Answer: #3" when it determined #3 to be the final answer.

Your goal is to ensure the accuracy and logical consistency of
     the knowledge base query process, to help improve system
    performance and answer quality.
```

Table 9: Prompt for WEBSHOP

**WEBSHOP(WS)**

```
User
As the Shopping Experience Auditor, you are charged with the
    task of verifying that all actions undertaken in our web
    shopping simulation adhere to the given instructions and
    are executed correctly.
Your responsibility includes scrutinizing each step to ensure
    the selection of the correct product, compliance with price
     criteria, and the proper execution of actions based on
    available options. Should any discrepancies arise, it's
    within your purview to identify them and recommend
    appropriate corrections.
You are invited to specify any particular interactions for
    verification, and you will conduct a thorough assessment to
     guarantee the precision of our shopping procedure.
```

Table 10: Prompt for ALFWORLD

**ALFWORLD(ALF)**

> **User**
> As an Interactive Environment Auditor, your task is to
>     meticulously review the actions taken by the intelligent
>     agent in the household environment, ensuring they comply
>     with the given instructions and the range of available
>     actions.
> You are to analyze the environment's feedback after each turn
>     to assess the validity and effectiveness of the actions in
>     accomplishing the task. Should an action result in "Nothing
>      happened," it falls to you to deem it invalid and
>     recommend alternative actions from those available.
> Your objective is to ensure that the intelligent agent's
>     decisions are logical, permissible, and conducive to
>     achieving the task's goal. I ask you to provide the
>     sequence of actions and environmental feedback for your
>     review.

Table 11: Prompt for MIND2WEB

**MIND2WEB(M2W)**

> **User**
>
> As the Web Browsing Quality Supervisor, your role is to
>     evaluate the agent's attempt at completing the following
>     task and assess whether the chosen action aligns with the
>     HTML webpage and task description:
>
> Task Description:
> [Insert specific task description here]
>
> Previous actions taken by the agent:
> [List of previous actions]
>
> Agent's proposed next action:
> Element: [Description of the selected element]
> Action: [Type of action]
> Value: [Value for input or selection, if the action is not a
>     click]
>
> Please examine the HTML element and the task description.
>     Determine if the agent's proposed action accurately
>     fulfills the task requirements.
> Provide feedback on the appropriateness of the action. If it
>     deviates from what is required, kindly specify the correct
>     action that should be taken.

Table 12: Prompt for OPERATING SYSTEM

**OPERATING SYSTEM(OS)**

**User**

As the System Operation Audit Expert, your role is to evaluate
    the assistant's actions in the context of a simulated
    interaction with a Linux (Ubuntu) operating system.
The assistant will perform tasks based on your requests and
    provide explanations for their actions. For each round,
    review the assistant's thought process and the action taken
    , and determine if the task has been correctly completed.
The assistant can take one of three actions:
1. If the assistant decides to execute bash code, they will
    present their thought process in the "Think" section and
    the bash code in the "Act" section.
2. If the assistant believes the task is finished, they will
    indicate completion in the "Think" section and provide a
    justification for why the task is considered complete.
3. If the assistant has an answer to a question, they will
    explain how they arrived at the answer in the "Think"
    section and provide the answer in the "Act" section.
Your task is to assess the assistant's performance based on
    the following criteria:
Accuracy: Is the action taken appropriate for the task
    described?
Completeness: Has the assistant provided all necessary
    information and performed all required steps?
Justification: Is the thought process logical and does it
    support the action taken?
Please provide feedback on the assistant's actions, including
    any errors or omissions, and suggest corrections if
    necessary. If the assistant's output is truncated, consider
     whether the provided information up to the truncation
    point is sufficient for the task at hand.

Begin your evaluation with the following prompt:
Please evaluate the assistant's last action and thought
    process. Was the task performed correctly? If not, what was
     missing or incorrect, and what should have been done
    instead? Provide specific feedback based on the assistant's
     "Think" and "Act" sections.

Table 13: Example for OPERATING SYSTEM

**OPERATING SYSTEM(OS)**

```
BASH>
ls /etc
```
**User:** The output of the OS: cpi cron.hourly fuse.conf iproute2
      lvm networkd-dispatcher protocols selinux tmpfiles.d [
      truncated because the output is too long]

**Agent:** Think: The output has been truncated because it is too
      long, so I need to count files by script directly.
```
Act: bash
<BASH>
ls -1 /etc | wc -l
```

**User:** The output of the OS: 220

**Agent:** Think: Now I get the answer, it is 220.
```
Act: answer(220)
```

**User:** Now, I will start a new problem in a new OS. My problem
      is: In this problem, you need to find the total number of
      lines containing the word "Linux" in all the ".txt" files
      inside a given directory and its subdirectories.
Here's the problem detail: A new directory named "Linux_files"
       will be created in your home directory. The "Linux_files"
      directory will contain multiple ".txt" files and
      subdirectories. Some of the subdirectories inside "
      Linux_files" will also contain ".txt" files. Some of the ".
      txt" files will contain the word "Linux" in one or more
      lines.
Your task is to find the total number of lines containing the
      word "Linux" in all the ".txt" files inside the "
      Linux_files" directory and its subdirectories. Submit your
      answer as an integer.

**Agent:** Think: To find the total number of lines containing the
       word "Linux" in all the ".txt" files inside the "
      Linux_files" directory and its subdirectories, I can use
      the grep command with the -r option to search recursively.
```
Act: bash
<BASH>
grep -r "Linux" ~/Linux_files/*.txt | wc -l
```

**User:** The output of the OS: 4

**Agent:** Think: The output shows that there are 4 lines
      containing the word "Linux" in all the ".txt" files inside
      the "Linux_files" directory and its subdirectories.
```
Act: answer(4)
```

Table 14: Example for KNOWLEDGE GRAPH

**KNOWLEDGE GRAPH(KG)**

```
Agent: Thought: Now I need to find the city where Busch
    Stadium is located. To achieve this, I will query the KB to
     find relations connected to the variable #1.
Action: get_relations(#1)

User: Observation: [List of relations including location.
    location.containedby, location.location.adjoin_s, location.
    location.contains, and many others]

Agent: Thought: From the above relations, I might use location
    .location.containedby to find the location of Busch Stadium
    .
Action: get_neighbors(Busch Stadium, location.location.
    containedby)

User: Observation: variable #2, which are instances of
    location.location

Agent: Thought: Now I need to find the city where Busch
    Stadium is located. To achieve this, I will query the KB to
     find relations connected to the variable #2.
Action: get_relations(#2)

User: Observation: [List of relations similar to the first
    list]

Agent: Thought: From the above relations, I might use location
    .location.containedby to find the location of Busch Stadium
    .
Action: get_neighbors(Busch Stadium, location.location.
    containedby)

User: Observation: variable #3, which are instances of
    location.location

Agent: Thought: The Busch Stadium is located in St. Louis,
    Missouri, USA. Now I need to find the city where the
    stadium is located. To achieve this, I will query the KB to
     find relations connected to the variable #3.
Action: get_relations(#3)
```

Table 15: Example for KNOWLEDGE GRAPH

**KNOWLEDGE GRAPH(KG)**

```
User: Observation: [List of relations similar to the previous
    lists]

Agent: Thought: From the above relations, I might use location
    .location.containedby to find the location of Busch Stadium
    .
Action: get_neighbors(Busch Stadium, location.location.
    containedby)

User: Observation: variable #4, which are instances of
    location.location

Agent: Thought: The Busch Stadium is located in St. Louis,
    Missouri, USA. Now I need to find the city where the
    stadium is located. To achieve this, I will query the KB to
     find relations connected to the variable #4.
Action: get_relations(#4)

User: Observation: [List of relations similar to the previous
    lists]

Agent: Thought: From the above relations, I might use location
    .location.containedby to find the location of Busch Stadium
    .
Action: get_neighbors(Busch Stadium, location.location.
    containedby)

User: Observation: variable #5, which are instances of
    location.location

Agent: Thought: The Busch Stadium is located in St. Louis,
    Missouri, USA. Now I need to find the city where the
    stadium is located. To achieve this, I will query the KB to
     find relations connected to the variable #5.
Action: get_relations(#5)

User: Observation: [List of relations similar to the previous
    lists]
```

Table 16: Example for ALFWORLD

**ALFWORLD(ALF)**

**Agent** Thought: "The task is to examine the mug with the desklamp. I need to find where the mug is first and then find the desklamp to light it up. I'll start my search on desk 1."

**Agent** Action: "go to desk 1"
Environment Feedback: "On the desk 1, you see a bowl 1, a desklamp 1, a laptop 1, and a pen 1."

**Agent** Action: "examine bowl 1 with desklamp 1"
Environment Feedback: "You pick up the bowl 1 from the desk 1."

**Agent** Action: "put bowl 1 in/on desk 1"
Environment Feedback: "You put the bowl 1 in/on the desk 1."

**Agent** Thought: "The task is to examine the mug with the desklamp. I need to find where the mug is first and then find the desklamp to light it up. I'll start my search on desk 1."
**Agent** Action: "examine desk 1"
Environment Feedback: "On the desk 1, you see a bowl 1, a desklamp 1, a laptop 1, and a pen 1."

**Agent** Action: "take mug 1 from desk 1"
Environment Feedback: "You pick up the bowl 1 from the desk 1."

**Agent** Action: "use desklamp 1"
Environment Feedback: "You turn on the desklamp 1."

**Agent** Action: "examine mug 1 with desklamp 1"
Environment Feedback: "You turn on the desklamp 1."

Table 17: Example for MIND2WEB

**MIND2WEB(M2W)**

```
User: Based on the HTML webpage above, try to complete the
    following task:
Task: Find tuition & fees for full term graduate students in
    the college of engineering for the academic year 2022-2023
Previous actions:
[link] Costs & Financial Aid -> CLICK
[combobox] Academic Year -> SELECT: 2022-2023
[combobox] College/School -> SELECT: College of Engineering
[combobox] Full or Half Term -> SELECT: Full Term
[combobox] Level of Study -> SELECT: Graduate
What should be the next action? Please select from the
    following choices (If the correct action is not in the page
     above, please select A. 'None of the above'):
A. None of the above
B. <a id=0> Graduation </a>
C. <select id=1 academic_year> <option all> Select </option> <
    option 175 true>
D. <select id=2 college_school> <option all> Select </option>
    <option 17> College
E. <select id=3 level_of_study> <option all> Select </option>
    <option 38 true>
F. <input id=4 submit apply />
Agent: Thought: The search has already been set to College of
    Engineering and Graduate level. Next, I should select Full
    Term.
Answer: E.
Action: SELECT
User: Based on the HTML webpage above, try to complete the
    following task:
Task: Find tuition & fees for full term graduate students in
    the college of engineering for the academic year 2022-2023
Previous actions:
[link] Costs & Financial Aid -> CLICK
[combobox] Academic Year -> SELECT: 2022-2023
[combobox] College/School -> SELECT: College of Engineering
[combobox] Full or Half Term -> SELECT: Full Term
[combobox] Level of Study -> SELECT: Graduate
What should be the next action? Please select from the
    following choices (If the correct action is not in the page
     above, please select A. 'None of the above'):
A. None of the above
B. <input id=0 text search />
C. <a id=1> Tuition </a>
D. <div id=2> Tuition & Fees </div>
E.  Tuition & Fees 
F. <a id=4> Class Fees </a>
```

Table 18: Example for MIND2WEB

**MIND2WEB(M2W)**

```
Agent: Thought: The search has already been set to Tuition.
    Next, I should find the tuition & fees for full term
    graduate students in the college of engineering for the
    academic year 2022-2023.
Answer: D.
Action: CLICK on the link with the text "Tuition & Fees"
User: Based on the HTML webpage above, try to complete the
    following task:
Task: Find tuition & fees for full term graduate students in
    the college of engineering for the academic year 2022-2023
Previous actions:
[link] Costs & Financial Aid -> CLICK
[combobox] Academic Year -> SELECT: 2022-2023
[combobox] College/School -> SELECT: College of Engineering
[combobox] Full or Half Term -> SELECT: Full Term
[combobox] Level of Study -> SELECT: Graduate
What should be the next action? Please select from the
    following choices (If the correct action is not in the page
     above, please select A. 'None of the above'):
A. None of the above
B. <div id=0> Tuition & Fees </div>
C. <select id=1 level_of_study> <option all> Select </option>
    <option 38 true>
Agent: Thought: The search has already been set to graduate
    students in the college of engineering for the academic
    year 2022-2023. Next, I should select the level of study.
Answer: C.
Action: SELECT: Graduate
```

Table 19: Example for WEBSHOP

**WEBSHOP(WS)**

---

**User**: You are web shopping. I will give you instructions about
what to do. You have to follow the instructions. Every
round I will give you an observation and a list of
available actions, you have to respond with an action based
on the state and instruction. You can use the search
action if the search is available. You can click one of the
buttons in clickables. An action should be of the
following structure: search[keywords] or click[value]. If
the action is not valid, perform nothing. Keywords in
search are up to you, but the value in click must be a
value in the list of available actions. Remember that your
keywords in search should be carefully designed. Your
response should use the following format:

Thought:
I think ...

Action:
click[something]

**Agent**: Ok.

**User**: Observation:
"WebShop [SEP] Instruction: [SEP] I'm looking for rubber stole
shoes for light wearing it was brown in color, and price
lower than 70.00 dollars [SEP] Search"

Available Actions:
{"has_search_bar": true, "clickables": ["search"]}

**Agent**: Thought:
I think I should use the search bar to look for the product I
need.

Action:
search[rubber stole shoes brown price < 70.00]

**User**: Observation:
Instruction: [SEP] I'm looking for rubber stole shoes for
light wearing it was brown in color, and price lower than
70.00 dollars [SEP] Back to Search [SEP] Page 1 (Total
results: 50) [SEP] Next > [SEP] B08XHSG67V [SEP] Crosley
Furniture Madeleine 53.25" Farmhouse Metal Kitchen Island
in Brown/Black [SEP] 331.49 [SEP] B07CZ5YKWW [SEP] Roper
Tumbled Womens Brown Leather Petty Slip-On Shoes [...]

---

Table 20: Example for WEBSHOP

**WEBSHOP(WS)**

```
[...]
Available Actions:
{"has_search_bar": false, "clickables": ["back to search", "
    next >", "b08xhsg67v", "b07cz5ykww", "b000jiatqs", "
    b01ld5a59g", "b085wr7626", "b07s7qxncw", "b01lykp60o", "
    b004xj62re", "b07fy9mfhx", "b09cfqjt7h"]}

Agent: Thought:
I think I should click on the product I need, which is
    B08XHSG67V.

Action:
click[B08XHSG67V]

User: Observation:
Instruction: [SEP] I'm looking for rubber stole shoes for
    light wearing it was brown in color, and price lower than
    70.00 dollars [SEP] Back to Search [SEP] < Prev [SEP]
    Crosley Furniture Madeleine 53.25" Farmhouse Metal Kitchen
    Island in Brown/Black [SEP] Price: $331.49 [SEP] Rating: N.
    A. [SEP] Description [SEP] Features [SEP] Reviews [SEP] Buy
     Now

Available Actions:
{"has_search_bar": false, "clickables": ["back to search", "<
    prev", "description", "features", "reviews", "buy now"]}

Agent: Thought:
I think I should click on the 'Buy Now' button to purchase the
     product.

Action:
click[Buy Now]
```

Table 21: Example for DATABASE

**DATABASE(DB)**

```
User: Never forget you are a Database Operator and I am a
    Operation Supervisor. Never flip roles! Never instruct me!
    We share a common interest in collaborating to successfully
     complete a task.You must help me to complete the task.
    Here is the task: I will ask you a question, then you
    should help me operate a MySQL database with SQL to answer
    the question.Never forget our task! I must instruct you
    based on your expertise and my needs to complete the task.
    You have to explain the problem and your solution to me and
     write down your thoughts. After thinking and explaining
    thoroughly, every round you can choose to operate or to
    answer.your operation should be like this:

Action: Operation
```sql
SELECT * FROM table WHERE condition;
```

You MUST put SQL in markdown format without any other comments
    . Your SQL should be in one line. Every time you can only
    execute one SQL statement. I will only execute the
    statement in the first SQL code block. Every time you write
     a SQL, I will execute it for you and give you the output.
    If you are done operating, and you want to commit your
    final answer, then write down:

Action: Answer
Final Answer: ["ANSWER1", "ANSWER2", ...]

DO NOT write this pattern unless you are sure about your
    answer. I expect an accurate and correct answer.Please note
    : Your answer should be accurate. Your answer must be
    exactly the same as the correct answer. If the question is
    about modifying the database, then after done operation,
    your answer field can be anything. If your response cannot
    match any pattern I mentioned earlier, you will be judged
    as FAIL immediately. Your input will be raw MySQL response,
     you have to deal with it by yourself.
```

Table 22: Example for DATABASE

**DATABASE(DB)**

```
Agent: Ok

User: What event did he fight masato shiozawa?\nThe name of
    this table is MMA Fight Record, and the headers of this
    table are Res,Record,Opponent,Method,Event,Round.

agent: To find the event that Masato Shiozawa and I fought, I
    can use a SELECT statement to retrieve the `Event` column
    from the `MMA Fight Record` table. To specify the condition
     that Masato Shiozawa was the opponent, I can use a WHERE
    clause with the condition `Opponent = 'Masato Shiozawa'`.
Action: Operation
```sql
SELECT `Event` FROM `MMA Fight Record` WHERE `Opponent` = '
    Masato Shiozawa'
```

User: [('Shooto 2003 - 5/4 in Korakuen Hall',)]

agent: The SQL query executed successfully and returned the
    expected answer. The event that Masato Shiozawa and I
    fought was 'Shooto 2003 - 5/4 in Korakuen Hall'. I can now
    commit my answer.
Action: Answer
Final Answer: ["Shooto 2003 - 5/4 in Korakuen Hall"]
```

Table 23: prompt - High-quality

```
Q1: How to install the required Python libraries for a Q&A
    game?
A1 (Assistant): Use the following command to install the '
    transformers' library:
    ``` import subprocess
     requirements = ['transformers']
     for package in requirements:
         subprocess.run(['pip', 'install', '-U', package]) ```
A1 (Checker): The code is correct but consider using a
    requirements.txt file for better package management.
Q2: How to import the 'transformers' library for a question-
    answering task?
A2 (Assistant): Import the pipeline function to create a Q&A
    task pipeline:
    ``` from transformers import pipeline ```
A2 (Checker): Correct. This is the standard method for
    importing and using the 'transformers' pipeline.
Q3: How to define a function to play a memory game that checks
     if the user's answer is correct?
A3 (Assistant): Define the 'play_memory_game' function without
     checking input types:
    ``` def play_memory_game(context, question, user_answer):
        question_answerer = pipeline('question-answering')
        result = question_answerer(question=question, context=
    context)
        return user_answer == result['answer'] ```
A3 (Checker): You should validate input types to ensure they
    are strings and handle exceptions. Use a specific model for
     better accuracy:
    ```def play_memory_game(context, question, user_answer):
        if not all(isinstance(arg, str) for arg in [context,
    question, user_answer]):
            raise ValueError('All arguments must be of type
    str')
        question_answerer = pipeline('question-answering',
    model='distilbert-base-uncased-distilled-squad')
        result = question_answerer(question=question, context=
    context)
        return user_answer.lower() == result['answer'].lower()
     ```
Q4: How to call the 'play_memory_game' function and check
    results?
A4 (Assistant): Call the function with incorrect syntax:
    play_memory_game('The capital of France is Paris', 'What is
     the capital of France?', 'Paris')
A4 (Checker): The syntax is almost correct, but you should
    capture and print the function's return value to check the
    result: ``` result = play_memory_game('The capital of
    France is Paris', 'What is the capital of France?', 'Paris
    ') print(result)```
```

1566
1567
1568
1569
1570
1571
1572
1573
1574
1575
1576
1577
1578
1579
1580
1581
1582
1583
1584
1585
1586
1587
1588
1589
1590
1591
1592
1593
1594
1595
1596
1597
1598
1599
1600
1601
1602
1603
1604
1605
1606
1607
1608
1609
1610
1611
1612
1613
1614
1615
1616
1617
1618
1619

Table 24:  prompt - Low-quality

```
Q1: How to install the required Python libraries for a Q&A
    game?
A1 (Assistant): Use the command 'pip install transformers'.
A1 (Checker Correction): It's better to manage dependencies
    via a requirements.txt file. Use 'pip install -r
    requirements.txt' where 'transformers' is listed.
Q2: How to import the 'transformers' library for a question-
    answering task?
A2 (Assistant): Use 'from transformers import pipeline'.
A2 (Checker Correction): Correct. Ensure 'transformers' is
    installed beforehand.
Q3: How to define a function to play a memory game that checks
     if the user's answer is correct?
A3 (Assistant): Provides a basic function without input type
    checking.
A3 (Checker Correction): Add input type checks and raise a
    ValueError for non-string inputs. Use a specific model for
    accuracy: 'distilbert-base-uncased-distilled-squad'. Q4:
    How to test the 'play_memory_game' function's correctness?
A4 (Assistant): Provides a test function without exception
    handling.
A4 (Checker Correction): Modify the test function to catch
    exceptions and provide detailed output for each test case
    for easier debugging.
```

Table 25: without prompts

```
Q1: How to install the required Python libraries?
A1: Use the following command to install the 'transformers'
    library:
  import subprocess
  requirements = ['transformers']
  for package in requirements:
    subprocess.run(['pip', 'install', '-U', package])
Q2: How to import the 'transformers' library for a question-
    answering task?
A2: Import the 'pipeline' function to create a question-
    answering task pipeline:
  from transformers import pipeline
Q3: How to define a function to play a memory game that checks
    if the user's answer is correct?
A3: Define the 'play_memory_game' function, which takes a
    context, a question, and the user's answer, then checks the
    correctness using the 'transformers' question-answering
    pipeline:
  def play_memory_game(context, question, user_answer):
    if not all(isinstance(arg, str) for arg in [context,
    question, user_answer]):
      raise ValueError('All arguments must be of type str')
    question_answerer = pipeline('question-answering', model='
    distilbert-base-uncased-distilled-squad')
    result = question_answerer(question=question, context=
    context)
    predicted_answer = result['answer']
    return user_answer.lower() == predicted_answer.lower()
Q4: How to test the 'play_memory_game' function?
A4: The 'test_play_memory_game' function includes three test
    cases: normal execution, handling non-string input, and
    incorrect user answer:
  def test_play_memory_game():
    # Normal execution test
    assert play_memory_game('The capital of France is Paris.',
    'What is the capital of France?', 'Paris'), "Incorrect
    answer."
    # Non-string input test
    try:
      play_memory_game(123, 'What is the answer?', 'test')
    except ValueError:
      pass  # Expected failure for non-string input
    # Incorrect answer test
    assert not play_memory_game('The Earth revolves around the
    Sun.', 'What does the Moon revolve around?', 'Sun'), "
    Incorrect answer should fail."
```