# OpenReview forum: "CMAT: A Multi-Agent Collaboration Tuning Framework for Enhancing Small Language Models"
_ICLR.cc/2025/Workshop/AgenticAI — ICLR 2025 Workshop AgenticAI Oral_

### Official Review · Reviewer_bbzd · 2025-02-27
**Farily Great Job**

**Rating:** 7
**Confidence:** 3

**Review:**

This paper introduces CMAT, a novel framework for enhancing small language models (SLMs) through multi-agent collaboration, and demonstrates its effectiveness via the TinyAgent model family. Below is a detailed review addressing the paper's strengths, weaknesses, and potential impact:

Summary of Contributions
The authors present two key innovations:
1. TinyAgent Models: Compact LLMs (1.8B–7B parameters) fine-tuned on Qwen and CodeLlama, achieving performance rivaling GPT-3.5 in tasks like SQL generation and OS interaction.
2. CMAT Framework: A multi-agent system with User, Assistant (Actor), and Checker (Critic) roles that enables dynamic weight updates through environmental feedback and memory mechanisms.

The framework integrates Chain-of-Thought reasoning, ReAct prompting, and supervised fine-tuning with LoRA/P-Tuning, showing a 58% improvement over base models in database tasks and 23% higher accuracy than AgentLM-7B in operating system interactions.

Strengths
1. Innovative Architecture: CMAT's Actor-Critic design with dual-memory management (short-term context + long-term reflection) addresses critical gaps in LLM adaptability. The ablation study (Table 5) validates the necessity of both agent-specific and general instructions.
2. Efficiency Gains: TinyAgent-7B matches GPT-3.5's performance with 20x fewer parameters (Table 2), demonstrating the viability of SLMs in resource-constrained environments.
3. Rigorous Evaluation: The six-task benchmark (OS, DB, KG, ALFWorld, WebShop, M2W) provides a holistic assessment. The DB task analysis (Figure 3) effectively highlights the reflection mechanism's impact on error correction.

Weaknesses and Recommendations
1. Limited Real-World Validation: Experiments rely on simulated environments (e.g., WebShop). Testing on live systems or industry datasets would strengthen claims about real-world applicability.
2. Narrow Baseline Comparison: While TinyAgent outperforms CodeLlama and Qwen, comparisons with other SLM frameworks (e.g., Microsoft's Phi-3) are absent.
3. Opaque Dataset Construction: Section 4.1 mentions "self-collected methods" but lacks details about data sources, cleaning protocols, or ethical considerations for crowd-sourced components.

Suggested Improvements:
1. Add latency/power consumption metrics to emphasize deployment advantages
2. Include error analysis for edge cases (e.g., nested SQL queries)

---

### Official Review · Reviewer_PeYb · 2025-02-28
**Good work**

**Rating:** 8
**Confidence:** 4

**Review:**

Summary:
This paper proposes CMAT (Collaborative Multi-Agent Tuning), a framework designed to enhance small language models through multi-agent collaboration with specialized roles and environmental feedback mechanisms. The authors claim their TinyAgent-7B model can match GPT-3.5 performance despite having fewer parameters. Key contributions include: (1) a dynamic real-time memory update system, (2) a role-playing mechanism for task allocation, and (3) experimental evaluation across multiple agent tasks.

Strengths:
1. The paper is generally well-structured with clear explanations of the framework components.
2. The dual memory system approach (short-term and long-term) addresses a recognized limitation in current LLMs.
3. Experimental results across multiple domains (OS, DB, KG, ALF, WS, M2W) show performance improvements.
4. The ablation study effectively demonstrates the value of combining agent-specific and general instructions.

Weaknesses:
1. Results and Comparisons:
   * Some performance improvements vary significantly across tasks
   * The comparison with existing agent frameworks could be more comprehensive
   * Additional baselines would strengthen the evaluation of the framework's effectiveness

2. Technical Contributions:
   * The Actor-Critic methodology could benefit from more implementation details
   * Clearer distinction between novel components and adaptations from prior work would help
   * More analysis on how the various components interact would strengthen the paper

Overall, it is a good paper, and I recommend accepting it.

---

### Official Review · Reviewer_um3C · 2025-03-04
**CMAT: A Multi-Agent Collaboration Tuning Framework for Enhancing Small Language Models**

**Rating:** 6
**Confidence:** 4

**Review:**

The paper introduces CMAT, a multi-agent collaboration framework designed to enhance small language models by enabling adaptive learning through agent feedback and memory mechanisms. It presents TinyAgent-7B, which achieves performance comparable to GPT-3.5 despite its smaller size, demonstrating efficiency in database querying, OS interactions, and web-based tasks. However, its effectiveness depends on the base model and the quality of the prompts. An ablation study examining each agent's contribution and a detailed analysis of error propagation and mitigation strategies would further strengthen the work. Overall, the paper is well-written and makes a significant contribution toward improving SLMs, with future work needed to address scalability, prompt independence, and real-world applicability.

---

### Decision · Program_Chairs · 2025-03-05

Accept (Oral)